# The antibody response to SARS-CoV-2 infection persists over at least 8 months in symptomatic patients

Riccardo Levi[1,4], Leonardo Ubaldi[1,4], Chiara Pozzi [2], Giovanni Angelotti[2], Maria Teresa Sandri[1,2], Elena Azzolini [1,2], Michela Salvatici[2], Victor Savevski[2], Alberto Mantovani [1,2,3] & Maria Rescigno [1,2 ✉]

### Abstract

**Background** Persistence of antibodies to SARS-CoV-2 viral infection may depend on several factors and may be related to the severity of disease or to the different symptoms.
**Methods** We evaluated the antibody response to SARS-CoV-2 in personnel from 9 healthcare facilities and an international medical school and its association with individuals' characteristics and COVID-19 symptoms in an observational cohort study. We enrolled 4735 subjects (corresponding to 80% of all personnel) for three time points over a period of 8–10 months. For each participant, we determined the rate of antibody increase or decrease over time in relation to 93 features analyzed in univariate and multivariate analyses through a machine learning approach.
**Results** Here we show in individuals positive for IgG ($\geq$12 AU/mL) at the beginning of the study an increase [$p = 0.0002$] in antibody response in paucisymptomatic or symptomatic subjects, particularly with loss of taste or smell (anosmia/dysgeusia: OR 2.75, 95% CI 1.753 – 4.301), in a multivariate logistic regression analysis in the first three months. The antibody response persists for at least 8–10 months.
**Conclusions** SARS-CoV-2 infection induces a long lasting antibody response that increases in the first months, particularly in individuals with anosmia/dysgeusia. This may be linked to the lingering of SARS-CoV-2 in the olfactory bulb.

### Plain language summary

SARS-CoV-2 infection activates the body's immune system to fight off infection. This immune response results in the production of proteins in the blood that target the virus called antibodies. The extent and duration of this antibody response may be associated with the type of symptoms the infected person is experiencing. Here, we analyzed SARS-CoV-2 antibody levels in individuals with asymptomatic, mild symptomatic (paucisymptomatic) and symptomatic disease in relation to the type of symptoms. We find that the antibody response is higher in people with symptoms and increases in the first three months, particularly in individuals with loss of smell or taste. In all people with SARS-CoV-2 antibodies at the start of the study, levels in the blood last for at least 8–10 months. Hence, SARS-CoV-2 infection is characterized by a long lasting antibody response which may protect from subsequent infections.

[1] Department of Biomedical Sciences, Humanitas University, Via Rita Levi Montalcini 4, 20072 Pieve Emanuele Milano, Italy. [2] IRCCS Humanitas Research Hospital, Via Manzoni 56, 20089 Rozzano Milano, Italy. [3] The William Harvey Research Institute, Queen Mary University of London, London, UK. [4]These authors contributed equally: Riccardo Levi, Leonardo Ubaldi. ✉email: maria.rescigno@hunimed.eu

t is becoming clear that the antibody response to SARS-CoV-2 can last at least 6 months in symptomatic patients[1], but it seems to decline in asymptomatics[2], including children[3,4]. Similarly, a reduction of antibody response in asymptomatic individuals was shown in a study with a fewer number of individuals ($n = 37$)[5]. In another study with a small number of subjects ($n = 68$), both IgG and IgA were shown to remain remarkably high for a 4 month period of time[6]. The antibody response in COVID-19 patients is associated with the establishment of a memory B cell response which is higher at 6 months[1], however, it is not clear whether there are features that correlate with this sustained B cell response. In addition, the antibody response to the spike protein might last for at least 7 months, while that to nucleocapsid wanes over time[7] and this correlates with a reduction in the IFN-γ producing CD8 + T cell response to the nucleocapsid[8]. The quality of an antibody response is fundamental to avoid long-term effects of COVID-19, indeed individuals with low or waning antibody response are more subject to long-COVID[9]. Hence, it is imperative to investigate the association between the antibody response, its duration, and the type of symptoms. We previously showed that an anti-SARS-CoV-2 serological analysis allowed us to follow the diffusion of the virus within healthcare facilities in areas differently hit by the virus[10]. At 8–10 months of distance, we analyzed the duration of this antibody response and evaluated whether there were features correlating with maintenance, reduction, or increase of the antibody response. We found that SARS-CoV-2 antibody levels increase in symptomatic subjects (particularly in individuals with anosmia/dysgeusia) in the first 3 months of the study. Moreover, this antibody response lasts for at least 8–10 months in all people with SARS-CoV-2 antibodies at the start of the study.

## Methods

**Study design**. This observational cohort study aimed at determining the anti-SARS-CoV-2 IgG plasma levels in nearly 4000 employees of nine healthcare facilities and an international medical school located in northern Italy. It has been approved by the institutional review board of Istituto Clinico Humanitas (ICH) for all participating institutes (ID:1374 IgG COVID-19 Humanitas and registered on clinicaltrial.gov NCT04387929). We have adhered to the STROBE reporting guidelines for observational studies. The serological determination was offered to all the employees of the involved sites and the anticipated refusal rate was assumed to be 10–15%. The overall IgG positivity was assumed to be around 10–15%. The primary endpoint was the number of test positive subjects. Given the study size, the study was able to estimate the overall positivity with a width of 95% confidence interval equal to 2% and the positivity for subgroups of at least 200 patients with a width of 95% confidence interval equal to 9%. No power analysis was performed to calculate the sample size. No randomization was performed. Accrual was on a voluntary basis (individuals aged ≥18 years; Humanitas group employees): it started on April 28th 2020 and more than 80% of personnel participated ($n = 4735$). The study foresees four blood collections every 3/4 months. Ten different centers participate: ICH, Rozzano (MI); Humanitas Gavazzeni, Bergamo; Humanitas Castelli, Bergamo; Humanitas Mater Domini (HMD), Castellanza (VA); Humanitas Medical Center (HMC), Varese; Humanitas University, Pieve Emanuele (MI); Humanitas San Pio X, Milano; Humanitas Cellini, Torino; Humanitas Gradenigo, Torino; Clinica Fornaca, Torino. This study was conducted following the Helsinki principles of good clinical practice and all participants signed informed consent and filled an anamnestic questionnaire before blood collection. In order to be tested, subjects had to fill in the questionnaire. Only after filling the questionnaire in its entirety, would the individuals be scheduled for blood sampling.

We analyzed 93 features (72 categorical and 17 numerical and four temporal) including, age, sex, location, professional role, the time between sample collections, COVID-19 symptoms (fever, sore throat, cough, muscle pain, asthenia, anosmia/dysgeusia (loss of smell and taste), gastrointestinal symptoms, conjunctivitis, dyspnea, chest pain, tachycardia, and pneumonia), home exits and smart-working, comorbidities (diabetes, asthma, neoplasia, autoimmunity, cardiovascular disorders, and hepatic disorders). We considered "asymptomatics" subjects without any symptoms; "paucisymptomatics" individuals that developed 1 or 2 symptoms; "symptomatics" individuals with three or more than three symptoms. None of the participants were enrolled at the time of symptoms. Thus, when the serological test was performed, they were either asymptomatics or the symptoms had disappeared. After excluding for employees that became positive for SARS-CoV-2 IgG ($n = 2$) during the observation period and those that dropped from phase 1 or for which we were missing at least two features, we analyzed 4534 participants (4.25% drop out) that participated to both phase 1 (April–May 2020) and phase 2 (July–August 2020). Among these, 499 subjects participated also to phase 3 (November–December 2020).

**IgG measure**. For the determination of IgG anti-SARS-CoV-2, the Liaison SARS-CoV-2 S1/S2 IgG assay (DiaSorin, Saluggia (VC), Italy) was used[11]. The method is an indirect chemiluminescence immunoassay for the determination of anti-S1 and anti-S2 specific antibodies. According to kit manufacturer, the test discriminates among negative (<15 AU/mL; with 3.8 as the limit of IgG detection) and positive (≥15 AU/mL) subjects. We considered positive subjects with IgG plasma levels ≥12 AU/mL rather than those with IgG ≥15 AU/mL, as suggested by the test manufacturer, based on our previous publication showing that these two groups behaved very similarly[10]. In addition, we considered also individuals with IgG comprised between 3.8 and 12 AU/mL (which we called IgG med: 3.8 < IgG <12 AU/mL). Consistency and reproducibility of the antibody test in samples collected in the two time points was confirmed for a limited number of individuals ($n = 50$) displaying different degrees of IgG positivity. The LIAISON assay's performance in comparison to a microneutralization assay is shown in Bonelli et al[11]. The LIAISON serological S1/S2 assay can distinguish between neutralization positive and negative samples at cut-offs near 15 AU/mL, and additionally, the data indicate that 92% of the samples with >80 AU/mL had neutralization titers ≥1:80, while 87% of samples with >80 AU/mL had neutralization titers ≥1:160.

As the samples were analyzed in separate batches, we compared the test accuracy on 21 samples from the phase 1 with the detection kits of phase 1 and phase 2 and demonstrated that the tested IgG were almost over-imposable (Supplementary Fig. 3).

**Statistical analysis and model**. We first cleared the dataset by eliminating data from all of those subjects that did not develop an IgG response over time (IgG ≤3.8 at the beginning and at the end of the examination) ($n = 2981$). We then analyzed the rate of antibody response defined as:

$$\text{RATE} = \frac{\text{IgG phase 2} - \text{IgG phase 1}}{\Delta \text{days}} = [\text{AU/mL} * \text{day}]$$

Positive rates mean increased antibody response, while negative rates indicate reduction of antibody response between the two analyzed time points.

For statistical analysis, we performed both a univariate and multivariate analysis. We applied Wilcoxon–Mann–Whitney statistical nonparametric test to compare the antibody rate distribution between classes of subjects (Tables 1 and 2).

**Table 1 Demographic distribution of antibody rates.**

| | | counts | min | 25 perc | 50 perc | 75 perc | max | mean | St.dev. | p value[a] |
|---|---|---|---|---|---|---|---|---|---|---|
| Sex | F | 1105 | −3.39 | −0.04 | −0.02 | 0.02 | 16.32 | 0.07 | 0.61 | 0.0139 |
| | M | 448 | −0.88 | −0.04 | −0.02 | 0 | 10.37 | 0.03 | 0.53 | |
| Age | 21–30 | 300 | −3.39 | −0.04 | −0.02 | 0.01 | 4.51 | 0.03 | 0.41 | 0.2644 |
| | 31–40 | 365 | −0.92 | −0.04 | −0.02 | 0.01 | 2.26 | 0.02 | 0.21 | 0.2302 |
| | 41–50 | 455 | −2.49 | −0.04 | −0.02 | 0.03 | 3.11 | 0.06 | 0.37 | 0.1083 |
| | 51–60 | 309 | −0.88 | −0.04 | −0.01 | 0.02 | 16.32 | 0.1 | 0.96 | 0.1442 |
| | 60+ | 124 | −0.68 | −0.04 | −0.02 | −0.01 | 10.37 | 0.12 | 0.99 | 0.0586 |
| BMI[b] | 18.5≤ BMI <25 | 940 | −0.92 | −0.04 | −0.02 | 0.01 | 10.37 | 0.03 | 0.41 | 0.3821 |
| | BMI ≥30 | 106 | −3.39 | −0.05 | −0.02 | 0.14 | 16.32 | 0.23 | 1.69 | 0.2182 |
| | 25≤ BMI <30 | 347 | −0.57 | −0.04 | −0.02 | 0.01 | 3.11 | 0.05 | 0.29 | 0.3933 |
| | BMI <18.5 | 73 | −0.22 | −0.04 | −0.02 | 0.02 | 1.27 | 0.05 | 0.26 | 0.3959 |
| IgG class phase 1 | IgG ≥12 | 613 | −3.39 | −0.08 | 0.02 | 0.23 | 16.32 | 0.18 | 0.92 | 2.1 E-10 |
| | IgG ≤3.8 | 74 | 0.01 | 0.01 | 0.02 | 0.03 | 0.08 | 0.02 | 0.01 | 1.5 E-15 |
| | 3.8 < IgG <12 | 866 | −0.13 | −0.03 | −0.02 | −0.01 | 0.83 | −0.02 | 0.04 | 8.0 E-22 |
| Role | Other[c] | 200 | −0.57 | −0.04 | −0.02 | −0.01 | 16.32 | 0.09 | 1.17 | 0.0116 |
| | Anesthesiologist | 19 | −0.83 | −0.04 | −0.02 | 0.01 | 0.09 | −0.05 | 0.2 | 0.4305[d] |
| | Biologist | 18 | −0.18 | −0.05 | −0.02 | −0.02 | 0.03 | −0.04 | 0.05 | 0.0978[d] |
| | Surgeon | 67 | −0.88 | −0.05 | −0.02 | 0.07 | 0.61 | 0.02 | 0.21 | 0.2653 |
| | Physiotherapist | 21 | −0.12 | −0.04 | −0.02 | 0.01 | 0.35 | 0.01 | 0.1 | 0.4204 |
| | Nurse | 398 | −3.39 | −0.04 | −0.02 | 0.04 | 3.11 | 0.08 | 0.41 | 0.0581 |
| | Physician | 210 | −0.92 | −0.04 | −0.01 | 0.02 | 10.37 | 0.08 | 0.75 | 0.0804 |
| | Healthcare partner operator | 149 | −2.49 | −0.03 | −0.01 | 0.17 | 1.55 | 0.1 | 0.38 | 0.0009 |
| | Front office (PARC) | 108 | −0.32 | −0.04 | −0.02 | 0.01 | 2.55 | 0.03 | 0.27 | 0.2892 |
| | Researcher | 50 | −1.5 | −0.03 | −0.02 | −0.01 | 4.51 | 0.06 | 0.68 | 0.1514 |
| | Cleaning service | 29 | −0.65 | −0.03 | −0.02 | 0.03 | 1.72 | 0.09 | 0.41 | 0.2414 |
| | Transport service | 14 | −0.39 | −0.04 | −0.01 | 0.02 | 2.22 | 0.13 | 0.62 | 0.4359[d] |
| | Staff | 188 | −0.4 | −0.04 | −0.02 | −0.01 | 0.98 | 0 | 0.15 | 0.0026 |
| | Student | 20 | −0.19 | −0.03 | −0.02 | −0.01 | 0.21 | −0.02 | 0.08 | 0.3705[d] |
| | Laboratory technician | 31 | −0.17 | −0.03 | −0.01 | −0.01 | 0.51 | 0.02 | 0.15 | 0.3243 |
| | Radiology technician | 31 | −0.18 | −0.03 | −0.02 | 0.01 | 1.01 | 0.04 | 0.22 | 0.4002 |
| Site | Other[c] | 21 | −0.19 | −0.04 | −0.02 | −0.01 | 2.22 | 0.08 | 0.49 | 0.2080 |
| | Casa di Cura Cellini | 51 | −0.17 | −0.05 | −0.03 | −0.01 | 1.27 | 0.01 | 0.22 | 0.0111 |
| | Clinica Fornaca di Sessant | 47 | −0.4 | −0.05 | −0.03 | 0 | 0.53 | 0 | 0.15 | 0.0833 |
| | Humanitas Castelli | 87 | −0.28 | −0.04 | 0.02 | 0.21 | 3.11 | 0.24 | 0.62 | 4.7 E-05 |
| | Humanitas Gavazzeni | 313 | −0.88 | −0.04 | −0.01 | 0.15 | 10.37 | 0.13 | 0.68 | 0.0001 |
| | Humanitas Gradenigo | 109 | −2.49 | −0.04 | −0.02 | −0.01 | 0.67 | −0.01 | 0.27 | 0.0586 |
| | Humanitas Mater Domini | 105 | −0.17 | −0.03 | −0.02 | −0.01 | 0.63 | 0 | 0.1 | 0.3412 |
| | Humanitas Medical Care | 23 | −0.19 | −0.04 | −0.02 | −0.01 | 0.02 | −0.03 | 0.04 | 0.0969 |
| | Humanitas Rozzano | 667 | −3.39 | −0.04 | −0.02 | 0 | 16.32 | 0.04 | 0.7 | 0.0338 |
| | Humanitas San Pio X | 98 | −0.68 | −0.03 | −0.02 | 0 | 2.39 | 0.04 | 0.29 | 0.2968 |
| | Humanitas University | 32 | −0.2 | −0.04 | −0.02 | −0.01 | 0.19 | −0.03 | 0.08 | 0.0590 |

[a]Wilcoxon–Mann–Whitney test.
[b]Some subjects did not indicate their BMI.
[c]Refers to volunteers and other professionals that operate in several structures.
[d]Minority class is less or equal to 20 (Wilcoxon–Mann–Whitney test is not reliable).

We analyzed the distribution of the rate feature and found a high value of kurtosis (461) around the median value of 0.016, hence to perform a multivariate analysis we restricted the data set to subjects with IgG rates either below the 10th percentile or above the 90th percentile to prevent a bias-variance problem in machine learning models and subjected the data to a linear regression analysis between the training and test data sets, where the target variable (rate of antibodies) was standardized using the Yeo–Johnson method[12]. We then applied Chi-squared statistical test to evaluate differences between classes and the rate thresholds described above (Tables 3 and 4). In order to evaluate the possible interactions between features and the rate of antibody response, we developed a multivariate approach to perform a binary classification between subjects who increased or decreased the level of antibodies. A set of seven logistic regressions has been applied on data using a bootstrap procedure (samples are drawn with replacement) and the output of each classifier has been averaged by a Bagging classifier to obtain the final output. The selection of hyperparameters of the machine learning model and the feature selection has been performed with a Bayesian optimization approach based on cross validation (fourfolds, stratified by outcome). The comparisons shown in Fig. 2 and Supplementary Fig. 2 were carried out using a one-tailed Wilcoxon matched-pairs signed-rank test. A probability value of $P < 0.05$ was considered significant. Data analyses were carried out using GraphPad Prism version 8 and Python version 3.8 with the following libraries: Pandas (version 1.1.4, data wrangling), Scipy (version 1.3.2, statistical analysis), Scikit-Learn (version 0.24.1, LR statistical model).

**Reporting summary**. Further information on research design is available in the Nature Research Reporting Summary linked to this article.

## Results

**Rate of the antibody response during phase 1 and phase 2 of SARS-CoV-2 diffusion.** We analyzed the persistence of the antibody response in healthcare workers that underwent

**Table 2 Antibody rates according to symptoms.**

| | | | counts | min | 25 perc | 50 perc | 75 perc | max | mean | St. dev | p value[a] |
|---|---|---|---|---|---|---|---|---|---|---|---|
| Class symptoms phase 1 (subjects with IgG ≥12) | Asymptomatic | | 91 | −1.5 | −0.13 | −0.05 | 0.09 | 2.39 | 0.04 | 0.5 | 0.00003 |
| | Paucisymptomatic | | 203 | −3.39 | −0.08 | 0.02 | 0.21 | 4.51 | 0.14 | 0.56 | 0.32865 |
| | Symptomatic | | 319 | −2.49 | −0.06 | 0.07 | 0.28 | 16.32 | 0.24 | 1.16 | 0.00057 |
| Symptoms phase 1 (subjects with IgG ≥12) | Fever | No | 350 | −3.39 | −0.09 | 0.01 | 0.21 | 4.51 | 0.13 | 0.02725 | 0.02725 |
| | | Yes | 263 | −2.49 | −0.06 | 0.05 | 0.25 | 16.32 | 0.24 | | |
| | Low-grade fever | No | 481 | −3.39 | −0.08 | 0.02 | 0.23 | 16.32 | 0.19 | 0.17265 | 0.17265 |
| | | Yes | 132 | −0.39 | −0.07 | 0.05 | 0.26 | 2.55 | 0.14 | | |
| | Cough | No | 372 | −3.39 | −0.08 | 0.01 | 0.19 | 16.32 | 0.14 | 0.01120 | 0.01120 |
| | | Yes | 241 | −2.49 | −0.07 | 0.07 | 0.31 | 10.37 | 0.24 | | |
| | Sore throat | No | 353 | −3.39 | −0.09 | 0.02 | 0.23 | 16.32 | 0.19 | 0.08309 | 0.08309 |
| | | Yes | 260 | −2.49 | −0.07 | 0.04 | 0.25 | 4.51 | 0.16 | | |
| | Muscle pain | No | 299 | −1.5 | −0.1 | 0 | 0.2 | 4.51 | 0.13 | 0.00763 | 0.00763 |
| | | Yes | 314 | −3.39 | −0.06 | 0.06 | 0.27 | 16.32 | 0.22 | | |
| | Asthenia | No | 341 | −3.39 | −0.1 | 0 | 0.21 | 16.32 | 0.17 | 0.00574 | 0.00574 |
| | | Yes | 272 | −2.49 | −0.06 | 0.07 | 0.25 | 10.37 | 0.19 | | |
| | Anosmia / dysgeusia | No | 313 | −3.39 | −0.12 | −0.01 | 0.2 | 16.32 | 0.14 | 0.00006 | 0.00006 |
| | | Yes | 300 | −0.86 | −0.05 | 0.06 | 0.28 | 10.37 | 0.22 | | |
| | Gastrointestinal symptoms | No | 403 | −3.39 | −0.08 | 0.03 | 0.21 | 10.37 | 0.17 | 0.46477 | 0.46477 |
| | | Yes | 210 | −2.49 | −0.08 | 0.02 | 0.25 | 16.32 | 0.19 | | |
| | Conjunctivitis | No | 517 | −3.39 | −0.08 | 0.02 | 0.21 | 16.32 | 0.18 | 0.16050 | 0.16050 |
| | | Yes | 96 | −2.49 | −0.07 | 0.04 | 0.32 | 1.72 | 0.14 | | |
| | Dyspnea | No | 493 | −3.39 | −0.08 | 0.02 | 0.23 | 16.32 | 0.16 | 0.34700 | 0.34700 |
| | | Yes | 120 | −2.49 | −0.07 | 0.05 | 0.26 | 10.37 | 0.24 | | |
| | Chest pain | No | 502 | −3.39 | −0.08 | 0.01 | 0.24 | 10.37 | 0.15 | 0.08088 | 0.08088 |
| | | Yes | 111 | −0.39 | −0.04 | 0.07 | 0.22 | 16.32 | 0.3 | | |
| | Tachycardia | No | 512 | −3.39 | −0.08 | 0.01 | 0.21 | 4.51 | 0.13 | 0.02353 | 0.02353 |
| | | Yes | 101 | −2.49 | −0.06 | 0.08 | 0.32 | 16.32 | 0.4 | | |
| | Pneumonia | No | 568 | −3.39 | −0.08 | 0.02 | 0.22 | 16.32 | 0.15 | 0.18692 | 0.18692 |
| | | Yes | 45 | −0.88 | −0.06 | 0.06 | 0.38 | 10.37 | 0.51 | 1.69 | |

[a]Wilcoxon–Mann–Whitney test.

immunological surveillance for SARS-CoV-2 exposure and resulted positive for anti-Spike 1/2 IgG (IgG ≥12 AU/mL). Although the test manufacturer considers positive subjects above 15 AU/mL and equivocal those between 12 and 15 (AU/mL), based on our previous publication showing that these two groups behaved very similarly we considered positive everybody ≥12 AU/mL[10]. The accrual was on a voluntary basis and did not occur in the symptomatic phase (at around 43 ± 17 days from COVID-19 assessment when symptomatic) (n = 4735). We excluded all of the individuals that became positive over the course of the analysis (n = 2) and those that dropped from phase 1 or for which we were missing at least two features, so to focus only on those individuals that were exposed during the first wave of infection to evaluate the duration of the antibody response (n = 4534). We assessed the correlation of the rate of antibody increase or decrease with the different analyzed features for the first two time points of observation in subjects with IgG ≥12 AU/mL. In Tables 1 and 2 are reported the rates for individual classes of features with relative statistical analysis. As shown, females sustained the antibody response better than males (p = 0.01); similarly, nonmedical healthcare professionals (specifically, healthcare partner operators) had higher antibody rates (p = 0.0009). The levels of antibodies increased in hospitals located in the Bergamo area (Castelli and Gavazzeni p < 0.0001) (Table 1) which was more hit by COVID-19 (37–43% of individuals with IgG ≥12)[10]. More important, the IgG rate in individuals which were positive for IgG (IgG ≥12 AU/mL; n = 613) at the beginning of the study was increased (p < 0.000001) over time, and this increase was either minor in asymptomatics (n = 91, p = 0.00003) and paucisymptomatics (n = 203) or strong in symptomatics (n = 319, p = 0.0006) (Table 2). This may explain why individuals from hospitals in the Bergamo area or nonmedical healthcare

professionals had higher levels of antibodies as most of them suffered from symptomatic COVID-19 (59 and 73%, respectively). On the contrary, those that had an intermediate IgG titer (3.8 < IgG <12 AU/mL considered as negative) displayed a significant reduction in IgG rate (p < 0.000001) (Table 1). However, this population is considered as negative for SARS-CoV-2 IgG according to the manufacturer. Many symptoms, including fever, cough, muscle pain, asthenia, tachycardia, and anosmia/dysgeusia, correlated with an increase of antibodies in the first two time points of observation (Table 2).

**The rate of the antibody response depends on the symptoms.** As we noticed that the distribution of the rate feature presented a high value of kurtosis (see Methods) we restricted the data set to subjects with IgG rates either below the 10th percentile [<−0.033 (n = 454)] or above the 90th percentile [>0.005 (n = 445)] to prevent a bias-variance problem in machine learning models. The accuracy of these rates was confirmed by linear regression analysis. In Fig. 1a, b are shown the regression diagnostic plots of predicted values against residuals of training and test data according to the threshold (<−0.033 AU/mL*day and >0.005 AU/mL*day). In Table 3 is shown the Chi-squared analysis for the populations below or above the set threshold rates. We found that as for the previous analysis, males reduced the level of antibodies more than females, even though this difference was no longer statistically significant (p = 0.06). The levels of antibodies increased in hospitals located in the Bergamo area (Castelli and Gavazzeni: p = 0.0032 and p = 0.0005, respectively) which was more hit by COVID-19 (37–43% of individuals with IgG ≥12) and most of the individuals were symptomatic[10] while it decreased in Humanitas Rozzano (p = 0.0806) which was less heavily hit (10% of individuals with IgG ≥12) and had

**Table 3 Chi-squared analysis of groups <10th percentile and >90th percentile.**

|  |  | <10 perc | >90 perc | *p* value |
|---|---|---|---|---|
| Sex | F | 321 | 340 | 0.0627 |
|  | M | 133 | 105 |  |
| Age | 21–30 | 93 | 83 | 0.5429 |
|  | 31–40 | 108 | 94 | 0.3804 |
|  | 41–50 | 121 | 142 | 0.0970 |
|  | 51–60 | 88 | 98 | 0.3711 |
|  | 60+ | 44 | 28 | 0.0793 |
| BMI[a] | 18.5≤ BMI < 25 | 267 | 255 | 0.6963 |
|  | BMI ≥ 30 | 58 | 72 | 0.1750 |
|  | 25≤ BMI < 30 | 106 | 93 | 0.4214 |
|  | BMI < 18.5 | 23 | 25 | 0.8261 |
| Role | Other[b] | 63 | 37 | 0.0109 |
|  | Anesthesiologist | 6 | 7 | 0.9701 |
|  | Biologist | 5 | 2 | 0.4640 |
|  | Surgeon | 25 | 21 | 0.7007 |
|  | Physiotherapist | 6 | 7 | 0.9710 |
|  | Nurse | 113 | 132 | 0.1255 |
|  | Physician | 56 | 61 | 0.6082 |
|  | Healthcare Partner Operator | 38 | 64 | 0.0062 |
|  | Front office (PARC) | 31 | 28 | 0.8494 |
|  | Researcher | 13 | 9 | 0.5485 |
|  | Cleaning service | 7 | 9 | 0.7698 |
|  | Transport Service | 5 | 5 | 0.7747[c] |
|  | Staff | 69 | 44 | 0.0214 |
|  | Student | 5 | 4 | 0.9759[c] |
|  | Laboratory Technician | 6 | 7 | 0.9701 |
|  | Radiology Technician | 6 | 8 | 0.7587 |
| Site | Other[b] | 6 | 3 | 0.5223[c] |
|  | Casa di Cura Cellini | 19 | 8 | 0.0573 |
|  | Clinica Fornaca di Sessant | 18 | 12 | 0.3828 |
|  | Humanitas Castelli | 23 | 47 | 0.0032 |
|  | Humanitas Gavazzeni | 97 | 142 | 0.0005 |
|  | Humanitas Gradenigo | 36 | 22 | 0.0918 |
|  | Humanitas Mater Domini | 20 | 23 | 0.7041 |
|  | Humanitas Medical Care | 8 | 3 | 0.2379[c] |
|  | Humanitas Rozzano | 190 | 160 | 0.0806 |
|  | Humanitas San Pio X | 27 | 21 | 0.5025 |
|  | Humanitas University | 10 | 4 | 0.1905[c] |

[a]Some subjects did not indicate their BMI.
[b]Refers to volunteers and other professionals that operate in several structures.
[c]Minority class is less or equal to 5 (chi-square test is not reliable).

machine learning model is a Bagging classifier of seven logistic regression, which was evaluated both on a training (Accuracy = 76.26; ROC AUC = 76.30; Recall = 81.14) and a test dataset (Accuracy = 72.00; ROC AUC = 72.12; Recall = 81.08), where the training/test split is 80–20%, stratified by the outcome. Classification metrics on the training set and test set are comparable, which shows that the model does not present overfitting on training data. In Fig. 1c is shown the multivariate logistic regression analysis. We found that the increased rate was associated primarily with anosmia/dysgeusia (regression coefficient = 1.0, 95% CI 0.56–1.46) and with chest pain (regression coefficient = 0.84, 95% CI 0.24–1.44), while the decreased rate was associated to subjects with intermediate IgG (3.8 < IgG <12) (regression coefficient = −1.61, 95% CI −2.03 to −1.0), which may be related to a noise in the instrument testing, and with past neoplasia (regression coefficient = −1.38, 95% CI −2.4 to −0.37). Interestingly, 54% of subjects with chest pain also presented loss of smell/taste while only 22% of subjects with smell/taste dysfunction also had chest pain, suggesting that IgG increase in the symptomatic population is primarily linked to anosmia and dysgeusia (not shown). In Fig. 1d are shown the odds ratio relative to Fig. 1c, which for chest pain is 2.32 (95% CI 1.27–4.24), for anosmia/dysgeusia is 2.75 (95% CI 1.75–4.30), for subjects with intermediate IgG (3.8 < IgG <12) is 0.2 (95% CI 0.13–0.30), and for subjects with past neoplasia is 0.25 (95% CI 0.09–0.69). Overall, these results indicate that although many symptoms are associated with an increase of IgG abundance in the observation time, only anosmia/dysgeusia and chest pain are associated to a higher IgG rate in the multivariate logistic regression analysis. By contrast the population with past neoplasia or intermediate levels of IgG (3.8 < IgG <12 AU/mL) are the ones that display a reduction in IgG. However, the significance of the reduction in the detection of antibodies in subjects with intermediate levels of IgG remains to be investigated, as this population did not represent early infected individuals because they were all nasopharyngeal swab negative[10].

**The antibody response lasts for at least 8–10 months**. We then analyzed whether the antibody response was maintained over time in the third time point of analysis (*n* = 499) which was evaluated between November and December 2020 thus reaching an observation of 8–10 months. As shown in Fig. 2 and Supplementary Fig. 2 we observed that both symptomatic and paucisymptomatic individuals still displayed a higher level of antibodies, however, they did not increase between phase 2 and phase 3. By contrast, asymptomatic individuals did not increase their IgG levels over time.

**Discussion**
We analyzed the 8–10-month duration of an antibody response to SARS-CoV-2 in personnel from nine healthcare facilities and an international medical school (Humanitas University) in Northern Italy in areas differently hit by the virus[10]. We show that the antibody response is stable both in symptomatic and asymptomatic/paucisymptomatic individuals and is increased in females and in nonmedical healthcare professionals. Previously, it has been shown in a study conducted in the British population that the antibody response declines of nearly 22% in symptomatic individuals and of 64% in asymptomatic individuals[2]. However, this study was based on a prick qualitative test and thus the decline may be related to the sensitivity of the test. We also observed that the antibody response declined when we analyzed the group (3.8 < IgG < 12 AU/mL) with IgG between the limit of detection (3.8 AU/mL) and the threshold of positivity (IgG ≥12 AU/mL), as set by the manufacturer. Whether this is linked to a difference linked to the sensitivity of the test or to a real reduction in an antibody response that may or may not be

less symptomatic individuals[10] (Table 3). The rate decreased in asymptomatic (65% of subjects fell in the group <−0.033; *p* < 0.000001), remained constant in paucisymptomatic, and increased in symptomatic individuals (62% of subjects were in the group >0.005; *p* < 0.000001) (Table 4). Interestingly, among the different symptoms, fever, cough, muscle pain, asthenia, dyspnea, tachycardia, chest pain, and anosmia/dysgeusia all correlated with a higher number of individuals falling into the group with rate >0.005, indicating that these symptoms were strongly associated with sustained/increased antibody response (0.000001 < *p* < 0.05, Table 4 and Supplementary Fig. 1). Among these, anosmia/dysgeusia was associated with the highest percentage of subjects presenting with increased IgG rate (69%; *p* < 0.000001, Table 4 and Supplementary Fig. 1).

**Increase of antibody response in subjects with anosmia/dysgeusia or chest pain**. Having observed differences according to sex, role, and site, and since many symptoms are linked, we performed a multivariate statistical analysis based on a supervised machine learning classification approach (see Methods). Through a Bayesian hyperparameter optimization algorithm, we assessed that the best

**Table 4 Chi-squared analysis of groups <10th percentile and >90th percentile per symptoms.**

| Symptoms | | <10 perc | >90 perc | p value | % Yes <10 perc[a] | %Yes >90 perc[a] |
|---|---|---|---|---|---|---|
| Fever | No | 360 | 284 | 3.9E-07 | | |
| | Yes | 94 | 161 | | 37 | 63 |
| Low-grade fever | No | 388 | 359 | 0.0678 | | |
| | Yes | 66 | 86 | | 43 | 57 |
| Cough | No | 335 | 284 | 0.0016 | | |
| | Yes | 119 | 161 | | 43 | 58 |
| Sore throat | No | 302 | 271 | 0.0923 | | |
| | Yes | 152 | 174 | | 47 | 53 |
| Muscle pain | No | 310 | 246 | 0.0001 | | |
| | Yes | 144 | 199 | | 42 | 58 |
| Asthenia | No | 340 | 268 | 3.7E-06 | | |
| | Yes | 114 | 177 | | 39 | 61 |
| Anosmia/dysgeusia | No | 364 | 247 | 4.0E-15 | | |
| | Yes | 90 | 198 | | 31 | 69 |
| Gastrointestinal symptoms | No | 336 | 313 | 0.2485 | | |
| | Yes | 118 | 132 | | 47 | 53 |
| Conjunctivitis | No | 400 | 382 | 0.3633 | | |
| | Yes | 54 | 63 | | 46 | 54 |
| Dyspnea | No | 405 | 375 | 0.0370 | | |
| | Yes | 49 | 70 | | 41 | 59 |
| Chest pain | No | 415 | 367 | 0.0001 | | |
| | Yes | 39 | 78 | | 33 | 67 |
| Tachycardia | No | 406 | 371 | 0.0107 | | |
| | Yes | 48 | 74 | | 39 | 61 |
| Pneumonia | No | 440 | 420 | 0.0889 | | |
| | Yes | 14 | 25 | | 36 | 64 |
| Total of Symptoms in phase 1 | Asymptomatic | 150 | 80 | 3.4E-07 | 65 | 35 |
| | Paucisymptomatic | 179 | 158 | 0.2520 | 53 | 47 |
| | Symptomatic | 125 | 207 | 5.6E-09 | 38 | 62 |
| Comorbidities | | | | | | |
| Chronic obstructive bronchopneumopathy | No | 451 | 444 | 0.6305[b] | | |
| | Yes | 3 | 1 | | | |
| Asthma | No | 430 | 412 | 0.2408 | | |
| | Yes | 24 | 33 | | | |
| Dyslipidemia | No | 413 | 404 | 0.9834 | | |
| | Yes | 41 | 41 | | | |
| Past neoplasia | No | 431 | 438 | 0.0063 | | |
| | Yes | 23 | 7 | | | |
| Hypertension | No | 401 | 409 | 0.0915 | | |
| | Yes | 53 | 36 | | | |
| Past coronaropathies | No | 454 | 443 | 0.4702[b] | | |
| | Yes | 0 | 2 | | | |
| Atrial fibrillation | No | 450 | 441 | 0.7439[b] | | |
| | Yes | 4 | 4 | | | |
| Past stroke/ TIA | No | 452 | 445 | 0.4878[b] | | |
| | Yes | 2 | 0 | | | |
| Steatosis/cyrrosis | No | 448 | 444 | 0.1359[b] | | |
| | Yes | 6 | 1 | | | |
| Chronic kidney failure | No | 453 | 445 | 0.9920[b] | | |
| | Yes | 1 | 0 | | | |
| Other liver diseases | No | 452 | 441 | 0.6641[b] | | |
| | Yes | 2 | 4 | | | |
| Rheumatological diseases | No | 445 | 431 | 0.3715 | | |
| | Yes | 9 | 14 | | | |
| Other diseases of the immune system | No | 418 | 409 | 0.9726 | | |
| | Yes | 36 | 36 | | | |
| Diabetes mellitus | No | 452 | 443 | 0.6305[b] | | |
| | Yes | 2 | 2 | | | |
| Gotta | No | 452 | 445 | 0.4878[b] | | |
| | Yes | 2 | 0 | | | |

[a]Percentage of subjects with symptoms (Yes) per rate class.
[b]Minority class is less or equal to 5 (chi-square test is not reliable).

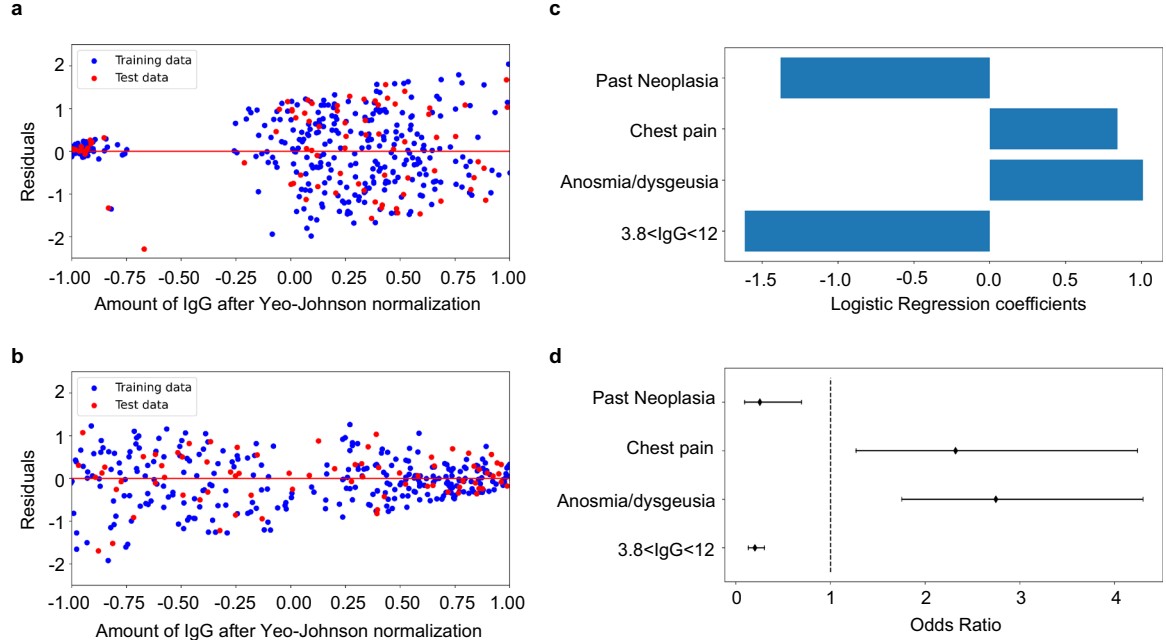

**Fig. 1 Results of linear and logistic regression for rate analysis in patients with >90th percentile and <10th percentile. a** Dataset <10th percentile regression diagnostic plot of amount of IgG after Yeo-Johnson normalization against residuals of training and test data ($n = 454$); **b** Dataset >90th percentile regression diagnostic plot of amount of IgG after Yeo-Johnson normalization against residuals of training and test data ($n = 445$); **c** Barplot with logistic regression coefficients for most important features; **d** Odds ratio of logistic regression with confidence intervals (95%) for the most important features.

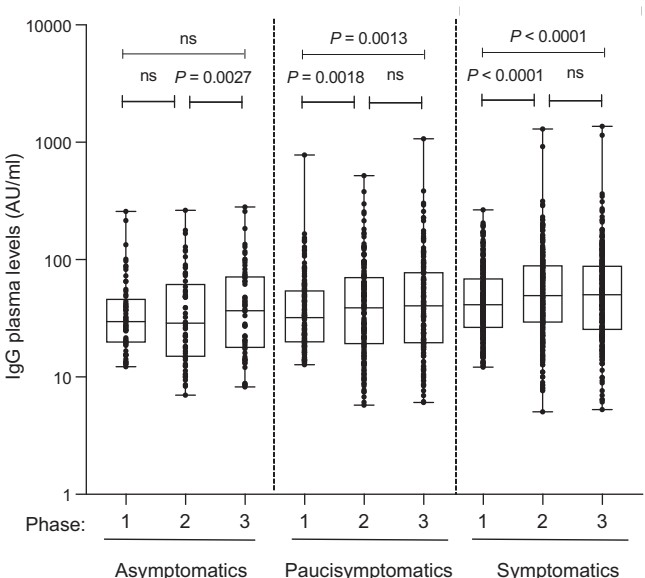

**Fig. 2 Anti-Spike S1/S2 IgG plasma levels.** Anti-Spike S1/S2 IgG plasma levels were measured in asymptomatics ($n = 61$), paucisymptomatics ($n = 163$), and symptomatics ($n = 275$) at three different time points (phase 1–3). Each dot corresponds to an individual subject. Log scale on the Y axis. The box plots show the interquartile range, the horizontal lines show the median values, and the whiskers indicate the minimum-to-maximum range. $P$ values were determined using a one-tailed Wilcoxon matched-pairs signed rank test.

specific to SARS-CoV-2, remains to be established. In a previous analysis, we excluded that this population represented individuals in the initial phases of an infection as all of them were tested for SARS-CoV-2 by a nasopharyngeal swab which however resulted negative[10]. When we analyzed the extremes, i.e., the individuals

with higher rates of antibody increase or decrease ($< -0.033$ and $>0.005$ AU/mL*day) we observed that asymptomatics had higher negative rates while symptomatics tended to continue increasing the antibody levels suggesting that extreme changes in rate separate the symptomatics from the asymptomatics. As during the observation time there was very limited viral diffusion in Northern Italy, as confirmed also by the finding that only two individuals became IgG positive and 2981 remained IgG negative throughout the study (all excluded from the analysis), we can conclude that the sustained or augmented antibody response may not be linked to reexposure to the virus.

In an attempt to address what improved the antibody response, we found that several symptoms were associated with increased rates of antibodies, however, in the multivariate logistic analysis, only anosmia/dysgeusia and chest pain were linked with the highest regression coefficients. Chest pain and anosmia are long-lasting symptoms in COVID-19 patients[13]. In addition, anosmia and/or dysgeusia are very common as they are found in around 50–70% of subjects affected by COVID-19[14,15]. In our cohort (Table 2), 49% of IgG-positive subjects had anosmia/dysgeusia, 28% chest pain, and 13.7% both anosmia/dysgeusia and chest pain, suggesting that indeed these two symptoms may, either alone or in combination, associate with IgG increase. We and others previously found that anosmia/dysgeusia together with fever were the symptoms that mostly characterized SARS-CoV-2 exposure[10,16]. In agreement, anosmia and dysgeusia have been proposed to be used to track SARS-CoV-2 diffusion[17]. Interestingly, SARS-CoV-2 can infect the olfactory epithelium[18,19], including olfactory sensory neurons, support cells, and immune cells, that express the viral entry receptors ACE2 and TMPRSS2[18,20,21]. Here, the virus can persist long and induce local inflammation[19] and olfactory bulb abnormalities[22–24]. In agreement, the loss of smell and taste can persist in individuals even with RT-PCR SARS-CoV-2 negativity in the nasopharyngeal swab for months[19,25]. Supporting this possibility, we did not detect any further increase of IgG levels between phase 2 and

phase 3 suggesting that when individuals eliminate the virus then there is no further increase of the antibody response.

One limitation of our study is that we followed our healthcare workers for the exposure to SARS-CoV-2 via measuring the anti S1/S2 IgG response and have not evaluated any other antibody subtype nor their neutralizing activity, even though the test used has correlated the antibody levels with their neutralizing activity, as reported in the Methods section.

Overall, these data suggest that increased antibody response in patients with anosmia/dysgeusia may be linked to the persistence of the virus in the olfactory bulb which through local inflammation and release of antigens, maintains and boosts the antibody response. This study opens new perspectives on the immunity to SARS-CoV-2 and warrants further investigation on the role of anosmia/dysgeusia on antibody response through the design of prospective observational studies coupling the testing of SARS-CoV-2 persistence in the olfactory bulb, loss of smell or taste and antibody titers. In addition, we show that the antibody response to the natural infection is durable and persists for at least 8 months. If the antibody response elicited by the vaccines is similarly effective, we may expect it to last for at least the same amount of time. Further, this observation strongly supports our findings and those of others that convalescent symptomatic COVID-19 patients should receive only one dose of vaccine[26–29] and suggests that this may occur even at months of distance from developing the disease as their antibody response will just need to be boosted.

## Data availability

The dataset is available at the link https://zenodo.org/record/5266408[30] with restricted license however available upon request. Patient informed consent does not allow for the deposition of clinical data in public access repositories. Interested researchers should contact biblioteca@humanitas.it to inquire about access; requests for noncommercial academic use will be considered and require ethics review. Source data for the main figures in the manuscript can be accessed as Supplementary Data 1.

## Code availability

Code developed for the analysis can be accessed at the link https://zenodo.org/record/5266426#.YSdIXN_ONaQ[31].

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

## Acknowledgements

This work was partially supported by a philantropic donation by Dolce & Gabbana, by the Italian Ministry of Health (Ricerca corrente) and by Fondazione Humanitas per la Ricerca. We would like to thank all the employees that volunteered to participate to this study, all the nurses and personnel that collected the samples, and the laboratory technicians that run the serological and rinopharyngeal tests. We would also like to thank the Humanitas management and staff, Drs Patrizia Meroni, Michele Lagioia, and Michele Tedeschi, who warmly supported this study for the safety of the employees.

## Author contributions

R.L. and L.U.: performed data analysis; C.P.: contributed to data analysis and manuscript writing; G.A. and V.S.: contributed to data analysis; M.T.S.: coordinated and supervised the laboratory analyses; E.A.: coordinated the recruitment and sampling of subjects (project administration) and participated in clinical study design; M.S.: carried out the laboratory analyses; A.M.: conceptualization and funding acquisition; M.R.: conceived the study, analyzed the data and wrote the manuscript.

## Competing interests

The authors declare no competing interests.
