## [Peer Review File · Communications Medicine]

Reviewers' comments:

Reviewer #1 (Remarks to the Author):

The manuscript from Levi et al reports on the antibody responses to SARS-CoV-2 from 4534 individuals followed by 5 months. Liaison SARS-CoV-2 assay was performed using S1/S2 to assess IgG levels. The results showed many clinical features, fever, cough, muscle pain, asthenia, tachycardia and anosmia/dysgeusia correlated with the increase of antibodies in the 5-month observation period for this cohort of individuals. All these symptoms correlated with sustained/increased antibody response. In general, this manuscript showed that the antibody response sustains in symptomatic and asymptomatic/paucisymptomatic individuals and is increased in females and in non-medical healthcare professionals in the cohort.

The strength of the manuscript is the characterization of the correlation of the rate of changes of the antibody responses with the different clinical features for 5 months duration. However, the data reported here is only a partial story, as the manuscript for some reason focuses only on the IgG responses but not the other immunoglobulin isotypes, e.g., IgM and IgA. Another caveat is that the current study has not tested the neutralization activity changes over time, which is of great importance for evaluating the antibody responses over time. At a minimum, these points and limitations should be clearly discussed.

Also, it would be better not to specifically refer anosmia/dysgeusia in the title.

Reviewer #2 (Remarks to the Author):

In the study by Levi et al., changes in anti-SARS-CoV-2 IgG antibody levels over time in personnel from different healthcare facilities in Italy are analyzed in the context of characteristics/symptoms of the study participants using a machine learning approach. Observations that increases in antibody titers may be associated with certain COVID-19 symptoms are interesting, but the study design and description of methods/results also raise several questions/concerns:

1. The DiaSorin brochure for the LIAISON SARS-CoV-2 S1/S2 IgG assay used in the study as well as a number of other publications using this assay list the following cut-off values for positivity:

- < 12 AU/mL = Negative
- 12 – 15 AU/mL = Equivocal
- > 15 AU/mL = Positive

In the current study the following values are used instead:

- < 3.8 AU/mL = Negative
- 3.8 – 12 AU/ml = IgG med
- > 12 AU/ml = Positive

More information on the choice of those values should be added. Is this a new recommendation by the manufacturer or a new convention based on a thorough validation? If not, the authors should follow the manufacturer's recommendation to analyze their data.

2. More information should be added on the recruitment of study participants. At what timepoint were participants enrolled in the study – e.g. were symptomatic individuals enrolled at a time when

they were still symptomatic or was the timepoint of enrollment random for both asymptomatic and symptomatic participants. The timepoint when serology was done is important for the assessment of changes in antibody titers. If serology was done early during the infection, I wouldn't be surprised about the higher antibody titer a few weeks/months later (especially in participants with more severe symptoms, who in general show substantial increases in IgG titers), because the first serology test might have been done before the peak in antibody titers expected at about 4 weeks after symptom onset. This peak might have been missed by testing participants early after symptom onset and then weeks later, so that antibody titers appear to be increasing but are actually decreasing if compared to the peak titer as reported in other publications.

3. In the methods section the authors describe that blood collections were done every 3 to 4 months. In the introduction and discussion, authors describe that the second timepoint for the antibody measurements was done 5 months later. The title of the manuscript suggests that antibody testing was done at a 5 months interval as well. The authors should clarify throughout the manuscript when the follow-up testing was done and if it was not 5 months later, they should change the title of the manuscript and all misleading statements throughout the text.

4. Were symptomatic study participants also tested with RT-PCR?

Were any of the participants who showed an intermediate IgG titer tested SARS-CoV-2 RT-PCR positive? Addition of this kind of information could help interpret the intermediate IgG group of participants and their decrease in antibody titers over time.

5. Were samples collected at enrollment and at the second timepoint tested in 2 separate batches or together at roughly the same time?

The authors mention in the methods that they confirmed consistency and reproducibility of the antibody test. These data could be added as a Figure panel or as level of agreement in the results text.

6. The authors may consider citing peer-reviewed publications instead of preprints (many preprints are cited for findings already published in peer-reviewed journals).

7. One suggestion (but no requirement) to considerably improve the manuscript is the addition of data for the next serology testing timepoint (just in case those data have become available in the meantime?) to validate findings with only one follow-up timepoint with a second follow-up.

8. The authors discuss the finding that anosmia and/or dysgeusia are linked to an increase in IgG between the two sampling/testing timepoints. Discussion of other factors that seem to be associated with an increase in titers should be added. It is difficult to explain why "being a non-medical healthcare professional" and "location of hospitals in certain areas" are associated with higher changes in antibody levels over time, especially because authors state that antibody responses are likely not linked to a re-exposure to the virus (due to the facts that only two study participants seroconverted during the study period and that spreading of the virus was reduced during that time).

Reviewer #3 (Remarks to the Author):

Summary: The authors analyzed the duration of antibody response against SARS-COV-2 and

evaluated whether there were features correlating with maintenance, reduction or increase of the antibody response in northern Italy. They identify anosmia/dysgeusia and chest pain to be associated with a higher IgG rate and those with neoplasia display a reduction in IgG. Authors then claim that increased antibody response in patients with anosmia/dysgeusia may be linked to persistence of the virus in the olfactory bulb which through local inflammation and release of antigens, maintains and boosts the antibody response.

- “Whether this is linked to a noise of the instrument that may change according to the test or to a real reduction in an antibody response that may or may not be specific to SARS-CoV-2, remains to be established.”
- Authors need to clarify how their serology assay and the associated instrument impacted their readings and results.
- “That indeed these two symptoms (chest pain and anosmia) may, either alone or in combination, associate with IgG increase.”
- Authors speculation that these two symptoms are linked to increase in IgG titers may need to be removed due to several confounders that are unknown. If the authors want to keep this claim, they need to elaborate and explain the mechanism and pathways that could be active for these associations.
- Statistical analysis seem sufficient, the machine learning model and its inferences can be described in more detail in the results.
- The impact of findings reported in this study on community and public health are not discussed. Nor are immediate value propositions for researchers and clinicians.

Reviewers' comments:

Reviewer #1 (Remarks to the Author):

The manuscript from Levi et al reports on the antibody responses to SARS-CoV-2 from 4534 individuals followed by 5 months. Liaison SARS-CoV-2 assay was performed using S1/S2 to assess IgG levels. The results showed many clinical features, fever, cough, muscle pain, asthenia, tachycardia and anosmia/dysgeusia correlated with the increase of antibodies in the 5-month observation period for this cohort of individuals. All these symptoms correlated with sustained/increased antibody response. In general, this manuscript showed that the antibody response sustains in symptomatic and asymptomatic/paucisymptomatic individuals and is increased in females and in non-medical healthcare professionals in the cohort.

The strength of the manuscript is the characterization of the correlation of the rate of changes of the antibody responses with the different clinical features for 5 months duration. However, the data reported here is only a partial story, as the manuscript for some reason focuses only on the IgG responses but not the other immunoglobulin isotypes, e.g., IgM and IgA. Another caveat is that the current study has not tested the neutralization activity changes over time, which is of great importance for evaluating the antibody responses over time. At a minimum, these points and limitations should be clearly discussed.

We thank the reviewer for this comment. This protocol was within the immunosurveillance of the hospital healthcare workers (>4000 individuals) of the Humanitas hospital group and hence we tested only IgG. We have added the following sentence in the discussion to highlight this limitation (Lines 173-176): *One limitation of our study is that we followed our healthcare workers for the exposure to SARS-CoV-2 via measuring the anti S1/S2 IgG response and have not evaluated other antibody subtypes nor their neutralizing activity even though the test used has correlated the antibody levels with their neutralizing activity as reported in the methods section (92% of the samples with >80 AU/mL had neutralization titers $\geq 1:80$).*

We have therefore added a sentence regarding the neutralization to the methods section (Lines 226-230). *'The LIAISON assay's performance in comparison to a microneutralization assay is shown in Bonelli et al.²². The LIAISON serological S1/S2 assay can distinguish between neutralization positive and negative samples at cut-offs near 15 AU/mL, and additionally the data indicate that 92% of the samples with >80 AU/mL had neutralization titers $\geq 1:80$, while 87% of samples with >80 AU/mL had neutralization titers $\geq 1:160$.'*

Also, it would be better not to specifically refer anosmia/dysgeusia in the title.

We changed the title to the following: "The antibody response to SARS-CoV-2 natural infection increases and persists over at least 8 months in symptomatic patients"

Reviewer #2 (Remarks to the Author):

In the study by Levi et al., changes in anti-SARS-CoV-2 IgG antibody levels over time in personnel from different healthcare facilities in Italy are analyzed in the context of characteristics/symptoms of the study participants using a machine learning approach. Observations that increases in antibody titers may be associated with certain COVID-19 symptoms are interesting, but the study design and description of methods/results also raise several questions/concerns:

1. The DiaSorin brochure for the LIAISON SARS-CoV-2 S1/S2 IgG assay used in the study as well as a number of other publications using this assay list the following cut-off values for positivity:

- < 12 AU/mL = Negative
- 12 – 15 AU/mL = Equivocal
- > 15 AU/mL = Positive

In the current study the following values are used instead:

- < 3.8 AU/mL = Negative
- 3.8 – 12 AU/ml = IgG med
- > 12 AU/ml = Positive

More information on the choice of those values should be added. Is this a new recommendation by the manufacturer or a new convention based on a thorough validation? If not, the authors should follow the manufacturer's recommendation to analyze their data.

In a previous study (Sandri MT, *et al.* SARS-CoV-2 serology in 4000 health care and administrative staff across seven sites in Lombardy, Italy. *medRxiv*, 2020.2005.2024.20111245 (2020) which we revised for Scientific Reports) we demonstrated that equivocal (12.0 – 15.0 AU/mL) and positive (>15.0 AU/mL) subjects behaved very similarly both as separate groups or as grouped together (Positive - IgG \geq 12.0 AU/ml). That is why we decided to consider “positive” subjects with IgG \geq 12.0 AU/ml. We considered “negative” people with IgG \leq 3.8 AU/ml that is the limit of IgG detection. Moreover, we hypothesized that there may be different exposures to the virus which result in a wide range of levels of antibody production (3.8 < IgG < 12 AU/ml).

We have clarified this in the methods: (Lines 220-223) “*We considered positive subjects with IgG plasma levels \geq 12.0 AU/mL rather than those with IgG \geq 15.0 AU/mL, as suggested by the test manufacturer, based on our previous publication showing that these two groups behaved very similarly⁴”.*

2. More information should be added on the recruitment of study participants. At what timepoint were participants enrolled in the study – e.g. were symptomatic individuals enrolled at a time when they were still symptomatic or was the timepoint of enrollment random for both asymptomatic and symptomatic participants. The timepoint when serology was done is important for the assessment of changes in antibody titers. If serology was done early during the infection, I wouldn't be surprised about the higher antibody titer

a few weeks/months later (especially in participants with more severe symptoms, who in general show substantial increases in IgG titers), because the first serology test might have been done before the peak in antibody titers expected at about 4 weeks after symptom onset. This peak might have been missed by testing participants early after symptom onset and then weeks later, so that antibody titers appear to be increasing but are actually decreasing if compared to the peak titer as reported in other publications.

None of the participants was enrolled at the time of symptoms. Thus, when the serological test was performed, they were either asymptomatics or the symptoms had disappeared. If they resulted positive for IgG they underwent a nasopharyngeal swab to exclude being actively infected. We analyzed the mean of serology date versus the appearance of COVID-19 symptoms and found that they were tested around 43 +/- 17 days after symptoms (see table and figure below). We have now clarified this in the text (Lines 56-61; 205-209).-

	Serological test - symptom onset (days)
count	315
mean	42,69
std	16,74
min	5
25%	35
50%	41
75%	48
max	111

3. In the methods section the authors describe that blood collections were done every 3 to 4 months. In the introduction and discussion, authors describe that the second timepoint for the antibody measurements was done 5 months later. The title of the manuscript suggests that antibody testing was done at a 5 months interval as well. The authors should clarify throughout the manuscript when the follow-up testing was done

and if it was not 5 months later, they should change the title of the manuscript and all misleading statements throughout the text.

Most of the individuals which suffered from COVID-19 (245 out of 315 of which we know the date of symptoms onset) had the disease in March 2020 whereas 26 people had the disease in February 2020. As we have included the third time point, which has become available in the meantime, and that was evaluated between November and December 2020, the duration of the analysis was between 8 and 10 months. Thus we have modified the text by saying 'at least 8 months'.

4. Were symptomatic study participants also tested with RT-PCR?

None of the participants was enrolled at the time of symptoms. Thus, when the serological test was performed, they were either asymptomatics or the symptoms had disappeared. However, we tested all subjects with $IgG \geq 12$ AU/ml in phase 1 and we excluded those subjects ($n=2$) that became positive during the observation period.

Were any of the participants who showed an intermediate IgG titer tested SARS-CoV-2 RT-PCR positive? In a previous analysis (Sandri MT, *et al.* SARS-CoV-2 serology in 4000 health care and administrative staff across seven sites in Lombardy, Italy. *medRxiv*, 2020.2005.2024.20111245 (2020)) a sample of 283 (100%) individuals with $3.8 < IgG < 12$ AU/mL underwent a nasopharyngeal swab for SARS-CoV-2 RNA viral detection and all of them resulted negative.

Addition of this kind of information could help interpret the intermediate IgG group of participants and their decrease in antibody titers over time.

We have added a sentence in the text at lines 146-148.

5. Were samples collected at enrollment and at the second timepoint tested in 2 separate batches or together at roughly the same time?

The samples were tested in 2 separate batches (3 considering also the third time point).

The authors mention in the methods that they confirmed consistency and reproducibility of the antibody test. These data could be added as a Figure panel or as level of agreement in the results text.

We have added a supplementary figure (Suppl. Fig. 1) to show the concordance of the two tests on 21 samples of phase 1.

6. The authors may consider citing peer-reviewed publications instead of preprints (many preprints are cited for findings already published in peer-reviewed journals).

We have substituted the reference that, in the meantime, have been peer-reviewed.

7. One suggestion (but no requirement) to considerably improve the manuscript is the addition of data for the next serology testing timepoint (just in case those data have become available in the meantime?) to validate findings with only one follow-up timepoint with a second follow-up.

In the meanwhile we had the opportunity to evaluate the response also of the third time point. We have added a new figure 2 and this has allowed us to reach an observation of 8-10 months. We observed that symptomatics still displayed a higher level of antibodies, however they did not increase between phase 2 and phase 3. By contrast, asymptomatic and paucisymptomatic individuals did not increase their IgG levels at any time (lines 126-131).

8. The authors discuss the finding that anosmia and/or dysgeusia are linked to an increase in IgG between the two sampling/testing timepoints. Discussion of other factors that seem to be associated with an increase in titers should be added. It is difficult to explain why "being a non-medical healthcare professional" and "location of hospitals in certain areas" are associated with higher changes in antibody levels over time, especially because authors state that antibody responses are likely not linked to a re-exposure to the virus (due to the facts that only two study participants seroconverted during the study period and that spreading of the virus was reduced during that time).

The Bergamo area was characterized by having more individuals with symptomatic infections (59% of subjects from Bergamo sites recorded more than 3 symptoms) and this may explain the difference in IgG levels. The same was true for non-medical healthcare (73% recorded more than 3 symptoms). We have added a sentence to explain this (Lines 72-74).

Reviewer #3 (Remarks to the Author):

Summary: The authors analyzed the duration of antibody response against SARS-COV-2 and evaluated whether there were features correlating with maintenance, reduction or increase of the antibody response in northern Italy. They identify anosmia/dysgeusia and chest pain to be associated with a higher IgG rate and those with neoplasia display a reduction in IgG. Authors then claim that increased antibody response in patients with anosmia/dysgeusia may be linked to persistence of the virus in the olfactory bulb which through local inflammation and release of antigens, maintains and boosts the antibody response.

“Whether this is linked to a noise of the instrument that may change according to the test or to a real reduction in an antibody response that may or may not be specific to SARS-CoV-2, remains to be established.”

- Authors need to clarify how their serology assay and the associated instrument impacted their readings and results.

This was referred only to the population with intermediate levels of IgG ($3.8 < \text{IgG} < 12 \text{ AU/mL}$). Now we have explained that this population is not considered as positive by the test and thus its significance remains to be investigated (Lines 123-125).

“That indeed these two symptoms (chest pain and anosmia) may, either alone or in combination, associate with IgG increase.”

- Authors speculation that these two symptoms are linked to increase in IgG titers may need to be removed due to several confounders that are unknown. If the authors want to keep this claim, they need to elaborate and explain the mechanism and pathways that could be active for these associations.

We have carried out a logistic regression analysis to take into account the different confounders and in the end of the discussion we have speculated why anosmia and dysgeusia may be related to increased antibody levels over time. Our speculations seem to be sustained by the third time point in which we do not observe anymore the correlation with anosmia and dysgeusia probably because the virus is finally eliminated.

- Statistical analysis seem sufficient, the machine learning model and its inferences can be described in more detail in the results.

We added some details at lines 102-107.

- The impact of findings reported in this study on community and public health are not discussed. Nor are immediate value propositions for researchers and clinicians.

We have added a sentence at the end of the discussion to highlight the implications for public health (Lines 182-188).

Reviewers' comments:

Reviewer #1 (Remarks to the Author):

The authors addressed all my concerns, I would recommend for publication.

For the ELISA cut-off issue raised by Reviewer 2, I would say, it is fine that the authors defined that in their own way, since there is no big difference.

Reviewer #2 (Remarks to the Author):

The authors have improved their manuscript and have responded to all of my previous comments.

I still think that the title of the manuscript, now being "The antibody response to SARS-CoV-2 natural infection increases and persists over at least 8 months in symptomatic patients" is not adequate because the authors have shown that antibody responses did not increase between their second and third observation time point. I would suggest changing the title to: "The antibody response to SARS-CoV-2 infection persists over at least 8 months in symptomatic patients"

In Figure 2 connecting lines of dots corresponding to the same individual should be shown for the three timepoints (if this looks very busy, the authors could add the plot with connecting lines to the supplement). It looks in the plot as if antibody responses are decreasing for most of the participants and only some had an increase in their titer.

Reviewer #4 (Remarks to the Author):

Preface: i will focus my comments on the computational analysis and the points raised by Rev 3 (as well as the author's response to these Rev3 comments).

Abstract:

the authors state in the abstract "This may be linked to the persistence of SARS-CoV-2 in the olfactory bulb." whereas in the title, the authors use the verb "persist" to describe the antibody response. This could be misleading. Please disambiguate these statements.

Results:

Lines 102-107:

- "We assessed that the best machine learning model is a Bagging classifier of 7 logistic regression"  how did you assess this? Did you compare different classifiers and then chose this one? If so, all these classifiers should have been evaluated using nested cross validation.
- Can you add the size/split of the training and test dataset?
- Fig 1A: there are two different point clouds. Can you further explain what these are?
- Fig 1: can you add to caption text and figures what the target prediction target was (antibody levels)
- can you show in the supplementary (or Fig 1) boxplots/violinplots [with all datapoints shown] where you plot Ab levels of those patients used for the regression model but stratified by Chest pain

(for example: high, low chest pain) as well as for the other prediction variables – so that the reader gets an idea of the differences? It would be nice to see this for all three patient groups. I know that these values are in the table provided – but tables are hard to parse and it would be informative to see the differences graphically.

Figure 2

- how was the statistical test performed? Did you perform multiple testing only within a patient group or also across patient groups? Since you are comparing three groups with one another, multiple testing needs to include all groups.
- From the medians, it seems like there is always an increase from time points 1 to 3, independently of patient group. I am unsure whether big claims can be made about this. It would be great if the authors can discuss this more.

Reviewers' comments:

Reviewer #1 (Remarks to the Author):

The authors addressed all my concerns, I would recommend for publication.

For the ELISA cut-off issue raised by Reviewer 2, I would say, it is fine that the authors defined that in their own way, since there is no big difference.

We thank the reviewer.

Reviewer #2 (Remarks to the Author):

The authors have improved their manuscript and have responded to all of my previous comments.

I still think that the title of the manuscript, now being "The antibody response to SARS-CoV-2 natural infection increases and persists over at least 8 months in symptomatic patients" is not adequate because the authors have shown that antibody responses did not increase between their second and third observation time point. I would suggest changing the title to: "The antibody response to SARS-CoV-2 infection persists over at least 8 months in symptomatic patients"

We thank the reviewer. We changed the title.

In Figure 2 connecting lines of dots corresponding to the same individual should be shown for the three timepoints (if this looks very busy, the authors could add the plot with connecting lines to the supplement). It looks in the plot as if antibody responses are decreasing for most of the participants and only some had an increase in their titer.

We added the Supplementary figure 2. In particular, to demonstrate that for most of the participants the antibody response is increasing we used a one-tailed test.

Reviewer #4 (Remarks to the Author):

Preface: i will focus my comments on the computational analysis and the points raised by Rev 3 (as well as the author's response to these Rev3 comments).

Abstract:

the authors state in the abstract "This may be linked to the persistence of SARS-CoV-2 in the olfactory

bulb." whereas in the title, the authors use the verb "persist" to describe the antibody response. This could be misleading. Please disambiguate these statements.

We rephrased the sentence in the abstract: "This may be linked to the lingering of SARS-CoV-2 in the olfactory bulb."

Results:

Lines 102-107:

- "We assessed that the best machine learning model is a Bagging classifier of 7 logistic regression"  how did you assess this? Did you compare different classifiers and then chose this one? If so, all these classifiers should have been evaluated using nested cross validation.

We thank the reviewer. We chose a Logistic regression model to obtain interpretable results of given covariates. Then, we applied a Bagging classifier approach to decrease the bias-variance problem (bootstrapping of the input data), as well as to increase accuracy performances. The number of Logistic regression used in the Bagging classifier has been defined after Bayesian hyperparameter optimization search (Methods section, lines 263-267 and main text, lines 106-108)

- Can you add the size/split of the training and test dataset?

We added these data in the main text (Line 110).

- Fig 1A: there are two different point clouds. Can you further explain what these are?

The statistical distribution of RATE variable higher than 90th quantile has an higher skewness (11.8) than the distribution for RATE variable less than 10th quantile (-9.6). After Yeo-Johnson normalization algorithm, the points close to the boundary remain close, whereas the other points spread on the right part of the graph.

- Fig 1: can you add to caption text and figures what the target prediction target was (antibody levels)

The prediction target is "Amount of IgG after Yeo -Johnson normalization".

We replaced "predicted values" with "Amount of IgG after Yeo -Johnson normalization" both in the figure and in the legend.

- can you show in the supplementary (or Fig 1) boxplots/violinplots [with all datapoints shown] where you plot Ab levels of those patients used for the regression model but stratified by Chest pain (for example: high, low chest pain) as well as for the other prediction variables – so that the reader gets an idea of the differences? It would be nice to see this for all three patient groups. I know that these values are in the table provided – but tables are hard to parse and it would be informative to see the differences graphically.

We added a Supplementary Figure 1 but unfortunately we could not divide in asymptomatic, paucisymptomatic and symptomatic because the number of subjects per group was reduced and we could not reach statistical significance for each individual feature.

Figure 2

- how was the statistical test performed? Did you perform multiple testing only within a patient group or also across patient groups? Since you are comparing three groups with one another, multiple testing needs to include all groups.

We thank the reviewer. Actually, we realized that the best statistical test to evaluate whether IgGs were increased after phase 1, was the one-tailed Wilcoxon matched-pairs signed rank test. We redid the analysis and we changed the figure 2. As is a paired analysis, we cannot compare all groups, but we have to perform a test only within a patient group (phase 1 vs phase 2 vs phase3 in asymptomatics or paucisymptomatics or symptomatics). We added also a supplementary figure 2 showing connecting lines of dots corresponding to the same individual.

- From the medians, it seems like there is always an increase from time points 1 to 3, independently of patient group. I am unsure whether big claims can be made about this. It would be great if the authors can discuss this more.

The reviewer is right regarding paucisymptomatic and symptomatic; for asymptomatic this is mostly due to a reduction in IgG between phase 1 and 2 and not to an increase between phase 1 and phase 3. It is possible that between phase 2 and 3 asymptomatics have been exposed again to the virus and this has led to an increase of IgG levels.

REVIEWERS' COMMENTS:

Reviewer #2 (Remarks to the Author):

All of my comments have been addressed.

Reviewer #4 (Remarks to the Author):

The authors have addressed all my concerns.